# METRIC SPACE MAGNITUDE FOR EVALUATING UNSUPERVISED REPRESENTATION LEARNING

## ABSTRACT

The *magnitude* of a metric space was recently established as a novel invariant, providing a measure of the 'effective size' of a space across multiple scales. By capturing both geometrical and topological properties of data, magnitude is poised to address challenges in unsupervised representation learning tasks. We formalise a novel notion of dissimilarity between magnitude functions of finite metric spaces and use them to derive a quality measure for dimensionality reduction tasks. Our measure is provably stable under perturbations of the data, can be efficiently calculated, and enables a rigorous multi-scale comparison of embeddings. We show the utility of our measure in an experimental suite that comprises different domains and tasks, including the comparison of data visualisations.

## 1 INTRODUCTION

Determining suitable low-dimensional representations of complex high-dimensional data is a challenging task in numerous applications. Whether its preprocessing biological datasets prior to their analysis (Nguyen & Holmes, 2019), the visualisation of complex structure in single-cell sequencing data (Lähnemann et al., 2020), or the comparison of different manifold representations (Barannikov et al., 2022): an understanding of structural (dis)similarities is crucial, especially in the context of datasets that are ever-increasing in size and dimensionality. The primary assumption driving such analyses is the *manifold hypothesis*, which assumes that data is a (noisy) subsample from some unknown manifold. Operating under this assumption, *manifold learning* methods have made large advances in detecting complex structures in data, but they typically use *local* measures of the embedding quality, which are ultimately relying on local approximations of manifolds by $k$-nearest neighbour graphs. However, such approximations—which require specific parameter choices and thresholds—can have a substantial negative impact on both embedding results and the interpretation of evaluation scores. Moreover, countering the increasing popularity of non-linear dimensionality reduction methods that claim to preserve local and global structures, recent work (Chari & Pachter, 2023) sheds some doubt on the assumption that 'good' embeddings should also faithfully preserve distances, while raising questions of how to measure the inevitable distortions introduced by representation learning. Thus, there is a need for novel methods in representation learning, which efficiently summarise data across varying levels of similarity, eliminating the need to rely on fixed neighbourhood graphs.

Motivated by these considerations, we adopt a more general perspective that does not rely on manifold approximations. To this end, we propose a novel embedding quality measure based on *metric space magnitude*, a recently-proposed mathematical invariant that encapsulates numerous important geometric characteristics of metric spaces. Specifically, magnitude summarises and captures characteristic properties like the diameter, curvature, density, and distance distribution of a space across varying scales of distances. Its computation relies on encoding dissimilarities between data points, permitting magnitude to capture relevant information at local and global scales. Magnitude is thus poised to compare metric spaces at different scales, yielding a compact holistic summary of embedding quality that is inherently aware of preserving geometric structure across multiple scales.

**Our contributions.** We develop a framework to formalise the notion of difference between the magnitude of two metric spaces in a scale-invariant manner. Moreover, we link our proposed magnitude difference to embedding quality and empirically demonstrate its utility both on simulated and real data, while also improving the computational efficiency of magnitude calculations. Finally, we prove the stability of magnitude and empirically validate its robustness to noise.

## 2 RELATED WORK

We briefly review the literature related to dimensionality reduction and previous applications of magnitude to machine learning.

**Dimensionality reduction.** Dimensionality reduction (DR) plays an important role in representation learning that affords many different uses, including (i) exploring data, (ii) extracting latent features, or (iii) compressing data (Nguyen & Holmes, 2019). It is thus of the utmost importance that DR results carry as much information about characteristic properties of the data as possible. A plethora of DR methods exist, including PCA, UMAP (McInnes et al., 2018), t-SNE (van der Maaten & Hinton, 2008), or methods based on (variational) autoencoders. The abundance of such techniques leads to the question of how to choose an appropriate method for a given task. The answer is often not straightforward, prompting practitioners to rely on embedding quality measures, i.e. scalar summaries designed to determine to what extent a specific structure or property of interest is preserved in lower dimensions. Numerous quality measures exist (Lee & Verleysen, 2009; Rieck & Leitte, 2017); they can be broadly split into two groups, depending on whether they measure the preservation of *local* or of *global* structures. However, when the goal is to preserve both of them, existing measures are not up to the task: neighbourhood-based measures like *trustworthiness* (Lee & Verleysen, 2008) fail to capture the global structures, whereas global measures like *distance correlation* (Székely et al., 2007) are unable to offer a sufficiently fine-grained view; as our experiments show, such measures are misleading even for toy examples like the 'Swiss Roll' dataset. Finally, existing methods are incapable to track properties across multiple scales, relying on fixed-scale neighbourhood graphs, for instance. There is thus a crucial need for a measure that can capture both the local and the global structure of a space across multiple scales.

**Magnitude and its applications in machine learning.** Magnitude has first been introduced by Solow & Polasky (1994) for measuring *biological diversity* in a set of species, with subsequent formalisations by Leinster (2013) in terms of category theory. Since then, the mathematical properties of magnitude have been studied more extensively, linking it to (i) maximum entropy, (ii) the *intrinsic dimension* of a space, or (iii) its *Euler characteristic*. Nevertheless, despite these strong ties to characteristic properties, magnitude has only rarely been applied in a machine learning context, and its potential for studying data remains under-explored. Recent publications started to bridge this gap. For instance, Bunch et al. (2021) successfully demonstrated the use of *magnitude weights* for boundary detection, while Adamer et al. (2021) showed how to use magnitude for *edge detection* in images. In the context of deep learning, a recent work links magnitude to the *generalisation error* of neural networks (Andreeva et al., 2023). However, somewhat surprisingly, we find that the question of how to compare the behaviour of metric space magnitude across two spaces has not yet been addressed—our work tackles this aspect in the context of dimensionality reduction.

## 3 BACKGROUND: THE MAGNITUDE OF A METRIC SPACE

*Magnitude* is a recently-developed invariant that describes the 'effective number of points' of a metric space as a function of its scaled distances (Leinster, 2013). We note that while we restrict our description to *finite* metric spaces, magnitude is defined for infinite spaces as well.

**Definition 3.1** (Magnitude of a metric space). *Let $X = \{x_1, \ldots, x_n\}$ be a finite metric space with an associated distance metric $d$. The* similarity matrix *of $X$ is defined as $\zeta_X(i,j) = e^{-d(i,j)}$ for $1 \leq i, j \leq n$. If $\zeta_X$ is invertible, the* magnitude *of $X$ is defined as*

$$\text{Mag}(X) := \sum_{ij} (\zeta_X^{-1})_{ij}, \qquad (1)$$

*while the* magnitude weighting vector $w_X$ *is defined as $w_X := \zeta_X^{-1}\mathbb{1}$. Denoting the $i$th element of $w_X$ by $w_{x_i}$, we obtain an equivalent characterisation of magnitude as $\text{Mag}(X) = \sum_i w_{x_i}$.*

The existence of magnitude is contingent on the existence of $\zeta_X$. Given a *negative definite metric $d$*, such as the $L_1$ and $L_2$ distance, $\zeta_X$ is invertible (Feragen et al., 2015). We will thus subsequently assume that $(X, d)$ with $X \subseteq \mathbb{R}^D$ permits the calculation of magnitude; in particular $X$ must *not* have any duplicate points.

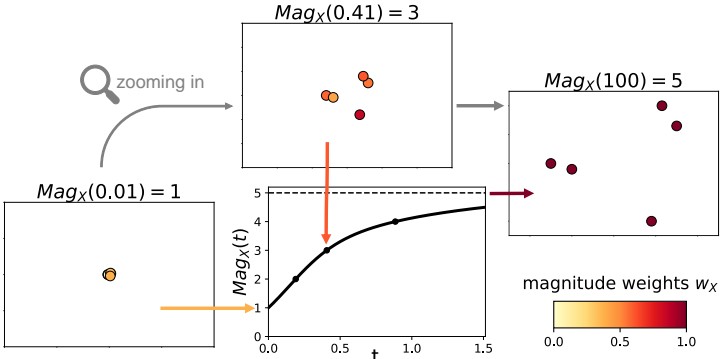

Figure 1: **Example of magnitude weights and the magnitude function for a metric space with 5 points.** When the scaling factor $t$ is very small, e.g. $t = 0.01$, the magnitude weights of all points sum up to approximately 1, so that magnitude is very close to 1, and the space effectively looks like one point. Following this, as we zoom in further, magnitude grows and at $t = 0.41$, 3 distinct clusters or points are visible. Finally, for $t = 100$, all the points are clearly separated, their magnitude weights converge to one, and the value of magnitude approaches 5, i.e. the cardinality of the space.

While the magnitude of a metric space is expressive even at a single scale (Leinster, 2013; Leinster & Shulman, 2021), magnitude unleashes its full potential in a multi-scale setting, assigning to a metric space not just a scalar but a function. To this end, we introduce a parameter $t \in \mathbb{R}_+$ and consider the metric space with distances scaled by $t$, which we denote by $tX$. Intuitively, this procedure corresponds to viewing the same space through different lenses, or at different 'zoom factors,' for example by converting distances from metres to centimetres. Computing the magnitude for each value of $t$ then yields the *magnitude function*.

**Definition 3.2** (Magnitude function). *For a metric space* $(X, d)$*, we define its scaled version* $tX := (X, d_t)$ *by* $d_t(x, y) := td(x, y)$*, where* $t \in \mathbb{R}_+$ *is the scaling factor. The* magnitude function *of* $(X, d)$ *is the function* $\mathrm{Mag}_X \colon t \mapsto \mathrm{Mag}(tX)$*.*

For $t \in (0, \infty)$, the magnitude function is defined for all but finitely many values of $t$ (Leinster, 2013). In our setting, given a negative definite metric $d$, the magnitude function is defined for *all* $t \in (0, \infty)$ because the scaled distance matrix is guaranteed to remain negative definite. For such metrics, the magnitude function is also *continuous* (Meckes, 2015, Corollary 5.5)[1] and for finite metric spaces, we have $\lim_{t \to \infty} \mathrm{Mag}(tX) = |X| = n$, i.e. the *cardinality* of $X$ (Leinster, 2013, Proposition 2.2.6). This limit behaviour exemplifies to what extent the magnitude function counts the 'effective number of points at scale $t$.'

## 4 METRIC SPACE MAGNITUDE TO ANALYSE EMBEDDINGS

Motivated by the characteristic properties of magnitude, we want to leverage it to assess the quality of embeddings, a common task in representation learning, which is often exacerbated by a lack of methods that are capable of comparing metric spaces. When assessing embedding quality, a dataset $X$ and its representation $Y$ can either be aligned, i.e. points in $X$ are in direct correspondence with points in $Y$, or unaligned. We will propose methods to address both scenarios, but our main goal is to compare *aligned* spaces, as these are more often encountered in the setting of dimensionality reduction. To address this setting, we define a novel method for comparing the magnitude functions of two (un)aligned spaces. We aspire towards a scale-invariant notion of distance between magnitude functions, because this property is necessary when accounting for the distortion of distances, which inevitably occurs under DR. As a first step towards scale invariance, we first normalise all distances by the diameter of the metric space, prior to calculating magnitude. This re-parametrisation of the magnitude function is permissible given the desired scale-invariance and the behaviour of magnitude under linear scaling as detailed in Corollary 4.2.

---

[1] Even for general metrics, the magnitude function is continuous for $t > t_{\mathrm{crit}}$, where $t_{\mathrm{crit}}$ is the supremum of its finitely many singularities.

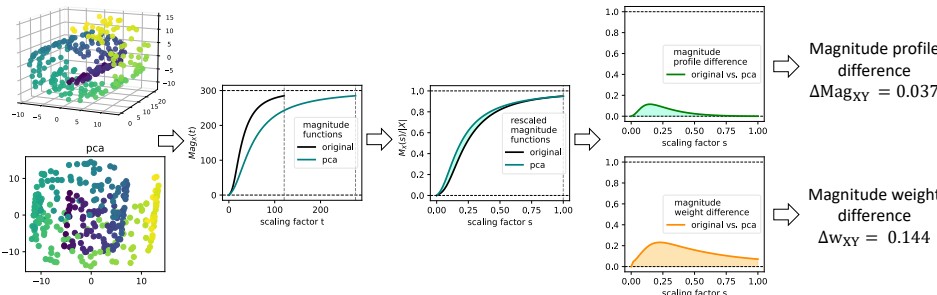

Figure 2: **Overview of the proposed dimensionality reduction quality measures.** Magnitude profile difference calculates the area between the re-scaled magnitude functions, which have been re-parametrised to the same approximate convergence point. Similarly, magnitude weight difference sums up the pairwise difference in magnitude weights across all re-scaled evaluation values. Lower values indicate a better agreement with the ground truth or original data.

### 4.1 COMPARING THE MAGNITUDE OF DIFFERENT METRIC SPACES

Existing work that computes magnitude functions for a specific space typically employs *fixed* evaluation intervals, thus possibly missing relevant scales. To address this limitation and to further incorporate knowledge on the distance distribution, we propose comparing magnitude across *multiple scales*, taking the overall convergence behaviour of magnitude functions into account. Our procedure automatically determines a suitable range of scaling factors, enabling us to compare magnitude functions in a *scale-invariant manner*. Intuitively, evaluation intervals should be related to the convergence behaviour of the magnitude function $\mathrm{Mag}_X(t)$, which converges to $|X|$ as $t \to \infty$. This convergence is monotonic in the sense that for $t$ sufficiently large, the magnitude function is increasing in $t$ under the condition that $tX$ is a *scattered space*, i.e. $t > \log(n-1)/\min_{x \neq y \in X} d(x,y)$ (Leinster, 2013), Given an error threshold $\epsilon > 0$, we thus argue that 'nothing interesting' happens after a certain scale $t_{\mathrm{conv}}$ that satisfies $\mathrm{Mag}_X(t_{\mathrm{conv}}) \geq |X| - \epsilon$, as we are only interested in the behaviour of magnitude until it starts converging. In practice, we set $\epsilon$ to a proportion of the cardinality, e.g $\epsilon = 0.05n$. We automate the process of finding the convergence point $t_{\mathrm{conv}}$ using numerical root-finding procedures. Next, having determined $t_{\mathrm{conv}}$, we evaluate magnitude at evenly-spaced intervals between $(0, t_{\mathrm{conv}}]$. This permits us to define the *re-scaled magnitude function*.

**Definition 4.1** (Re-scaled magnitude function). *Let $(X, d)$ be a metric space, $\mathrm{Mag}_X(t)$ its magnitude function, and $t_{\mathrm{conv}}$ its convergence point. For $s \in [0, 1]$, we then define the* re-scaled magnitude *function as $\mathcal{M}_X(s) := \mathrm{Mag}_X(s \cdot t_{\mathrm{conv}})$, and its re-scaled weights as $\mathcal{W}_X(s) = w_X(s \cdot t_{\mathrm{conv}})$.*

The re-scaled magnitude functions are advantageous because they permit us to reason about different metric spaces without having to account for (non)uniform metric scaling effects, making it possible to use the same domain for both functions. Subsequently, we will be operating solely with the *re-scaled magnitude functions* unless mentioned otherwise. Assuming that we have approximated the convergence scales $x_{\mathrm{conv}}$ and $y_{\mathrm{conv}}$ for two similar metric spaces $X$ and $Y$ of equal cardinality, we now want to compare them via magnitude. To this end, we propose the *magnitude profile difference*, a novel dissimilarity measure between the re-scaled magnitude functions $\mathcal{M}_X(s)$ and $\mathcal{M}_Y(s)$.

**Definition 4.2** (Magnitude profile difference). *Let $X$ and $Y$ be two metric spaces with $|X| = n_x$ and $|Y| = n_y$. Assume the associated (re-scaled) magnitude functions $\mathcal{M}_X(s)$ and $\mathcal{M}_Y(s)$ have been evaluated across the same scale factors $S = \{s_1, ..., s_i\}$. We define the point-wise magnitude profile difference to be $\Delta \mathrm{Mag}_{XY} := \int_0^1 |\frac{\mathcal{M}_X(s)}{n_x} - \frac{\mathcal{M}_Y(s)}{n_y}|ds \approx \sum_j |\frac{\mathcal{M}_X(s_j)}{n_x} - \frac{\mathcal{M}_Y(s_j)}{n_y}|.$*

The magnitude profile difference involves using the *area* between the curves as a measure of dissimilarity. The advantage of this formulation is that it affords an easy calculation while remaining interpretable. In practice, the summation from Definition 4.2 could also be replaced by more accurate numerical integration methods, such as Trapezoidal or Romberg integration. The difference between magnitude profiles now formalises the intuitive idea of comparing the behaviour of magnitude across two similar spaces as their magnitude approaches comparable convergence points. One can interpret this distance as comparing the behaviour of magnitude until each space effectively looks like it

has almost reached its cardinality. Alternatively, one may think about this comparison as starting with each space looking like one single point, then zooming in until the selected number of distinct observations are visible, and ultimately summing up the differences in magnitude at each zoom factor.

The magnitude profile difference studies the geometrical properties of each space across scales; it is thus even suitable for comparing unaligned metric spaces and spaces with different cardinalities. Nevertheless, whenever possible, tracking the identity of each point from the original to the reduced space is highly informative for assessing embedding quality. We thus propose another distance between $X$ and $Y$ based on the idea of *magnitude weights* from Definition 3.1.

**Definition 4.3** (Magnitude weight difference). *Let $X$ and $Y$ be two metric spaces with cardinality $n$ whose elements are aligned. Assume the associated re-scaled weight vectors $\mathcal{W}_X$ and $\mathcal{W}_Y$ have been evaluated across the same scaling factors $\{s_1, ..., s_i\}$. We define the magnitude weight difference function $\mathbf{w_{XY}} \colon j \in \{1, 2, .., i\} \to \mathbb{R}$ as $\mathbf{w_{XY}}(j) := \frac{1}{n} \sum_{k=1,2,..,n} |\mathcal{W}_X(s_j) - \mathcal{W}_Y(s_j)|_k$. Moreover, we define the magnitude weight difference between $\mathcal{W}_X$ and $\mathcal{W}_Y$ as $\Delta\mathbf{w_{XY}} := \sum_{j \in \{1,2,...,i\}} \mathbf{w_{XY}}(j)$.*

Notice that this definition of magnitude weight difference is similar to the previously-defined point-wise magnitude profile difference. Both notions assume that the magnitude values of $X$ and $Y$ evaluated at the scaling factors $s_j \cdot x_{conv}$ and $s_j \cdot y_{conv}$ respectively are comparable. The major distinction is that Definition 4.3 lends itself to a comparison on a per-point basis, whereas Definition 4.2 averages across points. We find the former to be beneficial in case data is paired or aligned (i.e. $x_i$ corresponds to $y_i$ under some transformation like an embedding) because it permits us to directly identify the changes in each point's magnitude weight before and after dimensionality reduction. We will thus use Definition 4.3 as the basis for an embedding quality measure.

**Computing magnitude.** A naïve calculation of magnitude according to Definition 3.1 requires inverting the similarity matrix $\zeta_X$, which has a worst-case complexity of $\mathcal{O}(n^3)$ and is numerically unstable. However, inverting $\zeta_X$ is not required in practice; instead, it suffices to solve certain *linear equations*. First, we notice that the calculation of magnitude can be written as $\mathrm{Mag}(X) := \mathbb{1}^\top \zeta_X^{-1} \mathbb{1}$. For finite metric spaces and negative definite metrics, $\zeta_X$ is a *symmetric positive definite matrix*, thus affording a *Cholesky decomposition*, which factorises $\zeta_X = LL^\top$, with $L$ being a *lower triangular matrix*. This operation is numerically stable and more efficient than matrix inversion (Higham, 2009). We thus have $\mathrm{Mag}(X) := \mathbb{1}^\top \zeta_X^{-1} \mathbb{1} = \mathbb{1}^\top (LL^\top)^{-1} \mathbb{1} = (L^{-1}\mathbb{1})^\top (L^{-1}\mathbb{1})$. This is equivalent to calculating $x^\top x$ with $x = L^{-1}\mathbb{1}$, which we can efficiently obtain by solving $Lx = \mathbb{1}$ since $L$ is lower triangular. Likewise, we can reformulate the calculating of the *magnitude weight vector* $w_X = \zeta_X^{-1}\mathbb{1}$ as solving $\zeta_X w_X = \mathbb{1}$, which also benefits from the Cholesky factorisation.

## 4.2 THEORETICAL PROPERTIES OF MAGNITUDE

Next to the theoretical properties linking magnitude to geometrical properties of a space, which we previously outlined, we prove that magnitude also satisfies properties that are advantageous in the setting of comparing different metric spaces. Specifically, we prove that magnitude is an *isometry invariant* and satisfies certain *stability properties* in light of perturbations of metric space.

**Definition 4.4** (Isometry). *Let $(X, d_X)$ and $(Y, d_Y)$ be two metric spaces. A map $f \colon X \to Y$ is called an* isometry*, or distance-preserving, if for any $a, b \in X$, we have $d_X(a, b) = d_Y(f(a), f(b))$. $X$ and $Y$ are called* isometric *if there is a* bijective isometry *from $X$ to $Y$.*

**Lemma 4.1** (Isometry invariance). *Given two isometric spaces $X, Y$, we have $\mathrm{Mag}_X = \mathrm{Mag}_Y$.*

**Corollary 4.1.** *The magnitude functions of two isometric spaces $X, Y$ are equal for all $t > 0$.*

Notice that the *converse* of this statement is not true in general, i.e. there are non-isometric spaces whose magnitude functions are the same (Leinster, 2013).

**Corollary 4.2.** *Let $X$ be a metric space and $Y = cX$ with $c \in \mathbb{R}_+$. Then the magnitude functions of $X$ and $\frac{1}{c}Y$ are equal. Also, the magnitude functions of $\frac{1}{\mathrm{diam}_X}X$ and $\frac{1}{\mathrm{diam}_Y}Y$ are equal, where $\mathrm{diam}_X := \max(d_X)$.*

Moreover, as a metric space invariant, magnitude should ideally satisfy certain stability properties. By this, we mean that if two metric spaces $X, Y$ are *close*, we want to obtain bounds on the differences

between their magnitude values. The canonical choice to measure closeness would be the Gromov–Hausdorff distance, but in the absence of strong results concerning the behaviour of magnitude under this distance (Leinster, 2013), we resort to a more general—but also weaker—notion of similarity in terms of *continuity*. More precisely, we will show that the similarity matrices used in the calculation of magnitude are well-behaved in the sense that closeness of metric spaces (under some matrix norm) translates to a continuous bound on the variation of the similarity matrices. We first prove a general result about matrices and their associated transformations.

**Lemma 4.2.** *Let* $\|A\|_2 := \sup\{\|Ax\|_2 : x \in \mathbb{R}^n \text{ with } \|x\|_2 = 1\}$ *refer to the* induced 2-norm for matrices*, and let* $A, B$ *be two* $n \times n$ *matrices with* $\|A - B\|_2 \leq \epsilon$. *Moreover, let* $f(M) := \mathbb{1}^\top M \mathbb{1}$. *Then* $\|f(A) - f(B)\|_2 \leq n\epsilon$.

Treating $A, B$ as inverse similarity matrices, the preceding statement shows that if the two inverse similarity matrices are close with respect to their spectral radius, the difference between their magnitude can be bounded. The following lemma shows that the similarity matrices satisfy a general continuity condition.[2]

**Lemma 4.3.** *Let* $(X, d_X)$ *and* $(Y, d_Y)$ *be two metric spaces with corresponding distance matrices* $D_X, D_Y$ *and cardinality* $n$. *For all* $\epsilon > 0$, *there exists* $\delta > 0$ *such that if* $|D_X - D_Y| < \delta$ *holds elementwise, then* $\|\zeta_X - \zeta_Y\|_2 \leq \epsilon$.

As a consequence of Lemma 4.3, and the continuity of matrix inversion, we know that magnitude is well-behaved under small perturbations of the respective distance matrices. Given a pre-defined threshold $\epsilon$, we can always find perturbations that preserve the magnitude difference accordingly. Notice that this result does not make any assumptions about the Gromov–Hausdorff distance of the metric space and only leverages the distance matrices themselves. Moreover, this result applies in case $X, Y$ are close with respect to the *Hausdorff distance*. If $d_H(X, Y) < \delta$, the elementwise condition $|D_X - D_Y| < \delta$ is satisfied *a fortiori*. Nevertheless, from a theoretical point of view, this result could be made stronger by showing bounds in terms of distances between the metric spaces. We leave such a result for future work, noting in passing that such strong results remain elusive at the moment (Govc & Hepworth, 2021); it is known, however, that the magnitude function is at least *lower semicontinous* (Meckes, 2013, Theorem 2.6).

## 5 EXPERIMENTS

In the following, we will demonstrate the utility of our *magnitude profile difference* and *magnitude weight difference* metrics in the context of evaluating DR methods. Using both simulated and real data, we demonstrate the following results: (i) The magnitude profile difference is empirically stable with respect to varying levels of Laplacian noise. (ii) In the context of non-linear DR, magnitude weight difference successfully identifies the ground truth and detects the underlying manifold. (iii) Magnitude difference relates local and global properties and finds a good compromise between cluster, distance, and density preservation for visualising the cluster structures of a subset of MNIST. (iv) When visualising single-cell data, magnitude difference identifies the information loss introduced by preprocessing. Further, for visualisation, it prefers representations that best reflect the underlying biological clustering structure while preserving the global geometry of the data.

### 5.1 EMPIRICAL STABILITY OF MAGNITUDE

We investigate the empirical stability of the magnitude profile difference. Given the difficulty in proving strong theoretical stability results, we verify that, in practice, the magnitude profile difference remains stable when adding noise to the input space. We thus sample points from a Laplace distribution with mean $\mu = 0$ and variance $2b^2$ with different levels of noise, i.e. $b \in \{0.0001, 0.001, 0.005, 0.01, 0.05\}$. Figure 3 depicts the errors in magnitude profile difference across three different datasets (circles, Swiss Roll, Gaussian blobs), using a different number of samples (varying between 100 and 5000 across 50 repetitions). The bound of 5000 points has been chosen given the clear downwards trend across multiple noise levels; we expect the same trend

---

[2]It is clear that the mapping itself is continuous because of the functions involved in its calculation. However, we find it important to remark on the bound obtained with respect to the *spectral norm* of the two similarity matrices.

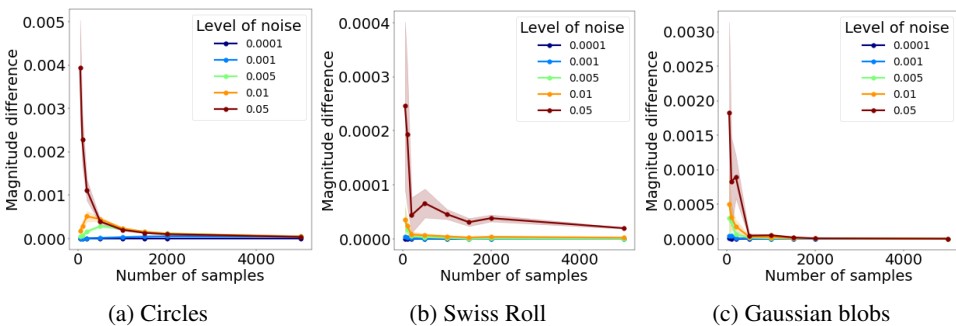

(a) Circles      (b) Swiss Roll      (c) Gaussian blobs

Figure 3: **Empirical stability of magnitude.** Magnitude difference is stable across different datasets and sample sizes. The lines show the mean magnitude difference and the shaded area the standard deviation calculated across 50 repetitions.

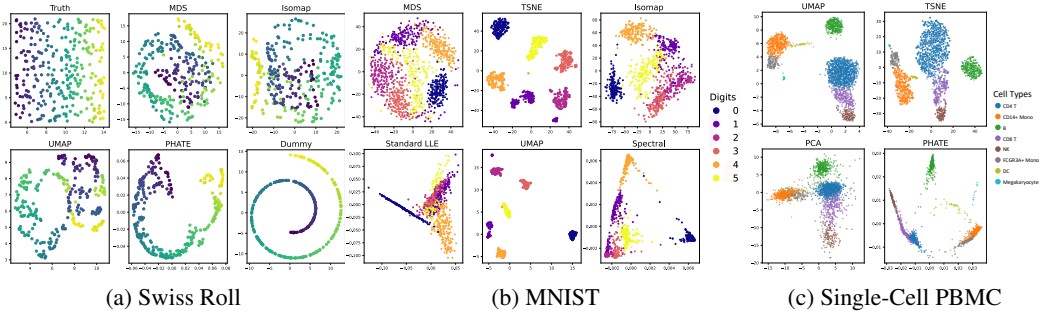

(a) Swiss Roll      (b) MNIST      (c) Single-Cell PBMC

Figure 4: **Examples of 2D embeddings** for three different datasets using various dimensionality reduction methods, ordered by their magnitude weight difference ($\Delta \mathbf{w_{XY}}$) from low (top left) to high (bottom right) values. As $\Delta \mathbf{w_{XY}}$ increases, there is a clearly-observable decrease in embedding quality and increase in distortion across these examples.

to hold for larger sample sizes. We observe that the magnitude profile difference does not increase above the value of $1 \times 10^{-4}$ with increasing sample size. In fact, the magnitude profile difference fluctuates more for smaller number of points, but this is still within a very small range. We therefore conclude that the magnitude profile difference between the original space and its noisy version does not change much, which indicates that our measure is reliable and stable across multiple experimental conditions.

## 5.2 COMPARING MANIFOLD LEARNING ALGORITHMS

To test the primary utility of our method, we analyse its behaviour in comparing embeddings arising from different manifold learning algorithms. As a simple toy example, we first compute the magnitude differences between a dataset with 300 points sampled randomly from a 3-dimensional 'Swiss Roll' and different two-dimensional representations, shown in Fig. 4(a) and ranked from best to worst. Magnitude weight difference clearly identifies the ground truth, which is given by the original points sampled on a 2D plane before being rolled up, as the best embedding. Magnitude properly detects the underlying manifold structure using only Euclidean distances. The reason for its success is that magnitude is more heavily influenced by close-by points rather than by the gap between the layers of the roll; magnitude is thus capable of detecting that 'zooming out' of the data is very similar to zooming out of the ideal planar representation. Meanwhile, other embedding quality measures based on the Euclidean distance, such as pairwise distance correlation, are observed to *fail* at this task. As summarised in Table 1, spectral distance correlation is the only other measure able to also detect the ground truth, but its calculation relies on the choice of $k$-NN graph, whereas magnitude weight difference arguably constitutes a more flexible, parameter-free multi-scale summary.

Table 1: **Comparison between magnitude weight difference and other quality measures**, with RMSE being calculated between the pairwise distance matrices, for the datasets and embeddings depicted in Fig. 4. We notice that $\Delta\mathbf{w_{XY}}$, our proposed magnitude weight difference, selects the ground truth for the Swiss Roll, determines t-SNE as a good method for visualising clusters in the MNIST digits data and prefers UMAP as a good compromise for the PBMC single-cell data.

| | $\Delta\mathbf{w_{XY}}$ ($\downarrow$) | $\Delta\mathrm{Mag_{XY}}$ ($\downarrow$) | Dist. corr. ($\uparrow$) | Spectral dist. corr. ($\uparrow$) | Density preservation ($\uparrow$) | RMSE ($\downarrow$) | Trustworthiness ($\uparrow$) | Silhouette Score ($\uparrow$) |
|---|---|---|---|---|---|---|---|---|
| | | | | SWISS ROLL | | | | |
| Ground Truth | 0.129 | 0.007 | 0.459 | 0.0813 | 0.409 | 8.6 | 0.80 | |
| MDS | 0.130 | 0.021 | 0.883 | −0.0006 | 0.520 | 3.2 | 0.93 | |
| Isomap | 0.146 | 0.028 | 0.844 | −0.0319 | 0.248 | 8.7 | 0.92 | |
| UMAP | 0.163 | 0.044 | 0.643 | 0.0348 | 0.408 | 12.4 | 0.98 | |
| PHATE | 0.172 | 0.075 | 0.406 | −0.0008 | 0.316 | 16.2 | 0.91 | |
| Dummy | 0.210 | 0.152 | 0.844 | −0.0105 | 0.426 | 4.4 | 0.89 | |
| | | | | MNIST DIGITS | | | | |
| MDS | 0.259 | 0.233 | 0.779 | 0.0638 | 0.583 | 15.3 | 0.91 | 0.30 |
| t-SNE | 0.289 | 0.267 | 0.571 | −0.0154 | 0.425 | 21.3 | 0.98 | 0.67 |
| Isomap | 0.312 | 0.285 | 0.649 | 0.0351 | 0.484 | 41.2 | 0.90 | 0.44 |
| LLE | 0.322 | 0.293 | 0.416 | 0.0747 | 0.144 | 49.3 | 0.82 | 0.08 |
| UMAP | 0.324 | 0.300 | 0.521 | −0.0025 | 0.127 | 37.7 | 0.98 | 0.74 |
| Spectral | 0.356 | 0.322 | 0.554 | −0.0008 | 0.430 | 49.3 | 0.92 | 0.55 |
| | | | | SINGLE-CELL PBMC | | | | |
| UMAP (40 PCs) | 0.390 | 0.379 | 0.331 | 0.0430 | 0.193 | 49.8 | 0.63 | 0.49 |
| tSNE (40 PCs) | 0.442 | 0.430 | 0.304 | 0.1040 | −0.233 | 28.9 | 0.64 | 0.40 |
| PCA (2 PCs) | 0.437 | 0.424 | 0.447 | 0.0700 | 0.407 | 48.1 | 0.61 | 0.41 |
| PHATE (40 PCs) | 0.454 | 0.440 | 0.334 | 0.0440 | 0.322 | 56.4 | 0.62 | 0.62 |

### 5.2.1 MNIST AND CLUSTERING

Motivated by the results on the 'Swiss Roll' dataset, we now analyse data that exhibits distinct clustering patterns. To this end, we consider a dataset of handwritten MNIST digits, which we further subset to $1,083$ greyscale images, corresponding to the digits from $0$ to $5$. We choose these data in order to visualise and visually assess the six distinct clusters more easily. From the two-dimensional representations shown in Fig. 4(b) and Fig. 7 (in the appendix), magnitude weight difference identifies MDS as the best embedding, which is likely because both MDS and magnitude assess embedding quality in terms of Euclidean distances. This is followed by t-SNE, which in this case is the representation that clearly separates different digits, exhibiting the highest silhouette score of $0.67$, the highest trustworthiness, the lowest neighbourhood loss, and the second-lowest RMSE as shown in Table 1 and Table 3 (in the appendix). Magnitude thus demonstrates its ability to take cluster structure into account, finding the best compromise between cluster preservation, preservation of local neighbourhoods, and preservation of all distances. By contrast, embeddings like PCA and Isomap that show separate groups but do not clearly separate between them, have lower magnitude differences than UMAP or PHATE, which exhibit distinct clusters but worse RMSE and distance correlation, because they introduce more global distortion. LLE and the spectral embedding generally are ranked worse in terms of magnitude and other measures, confirming the visual suspicion that they introduce spurious structures. Thus, since tracking the number of distinct points is related to clustering behaviour, we have shown that magnitude weight difference emphasises methods that preserve both individual groups and the relationship between them.

**Magnitude finds suitable embeddings.** We find that in comparison to other quality measures, magnitude weight difference is correlated with distance correlation ($r = -0.85$), RMSE ($r = 0.81$) and density preservation ($r = -0.79$), lending credence to the numerous beneficial geometrical and topological properties captured by magnitude. However, it still regards t-SNE as a suitable embedding, despite its comparatively low distance correlation, high local density preservation, and strong preservation of clusters. This highlights that magnitude weight difference offers a more holistic view of an embedding than existing measures: in fact, although magnitude is related to ideas such as distance preservation and density preservation, magnitude in fact does *not* measure the same structure as existing quality metrics, but finds a compromise between preserving the global and local geometry of data. Furthermore, while distance correlation is limited to comparing individual pairwise distances, and local density preservation is limited by its choice of static local neighbourhoods, the magnitude difference compares *all* distances simultaneously across multiple scales without the need to manually pick a neighbourhood size, constituting a parameter-free method (except for the choice of convergence point, which we argue can be made canonically).

### 5.2.2 SINGLE-CELL DATA ANALYSIS AND VISUALISATION

Finally, we assess the utility of magnitude in a single-cell data analysis context. Here, DR methods are commonly used for simplifying gene expression features prior to analysis or for explanatory data analysis. However, due to the complex nature of these data and challenges in assessing the loss of global, local, or distance-related information after dimensionality reduction, visualising and interpreting the resulting representations remains an important open problem (Lähnemann et al., 2020). Motivated by this, we use magnitude difference to assess embedding quality for a data set of human peripheral blood mononuclear cells (PBMCs), consisting of $2,638$ cells and $1,838$ features.

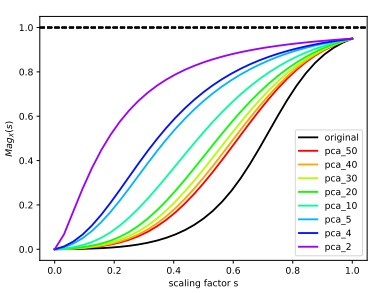

Figure 5: **Re-scaled magnitude functions of the single-cell data for varying numbers of principal components (PCs).** The difference to the original magnitude function serves as a reliable indicator of information loss and distortions as the number of PCs decreases.

When following the common practice of using PCA for initial pre-processing (Heumos et al., 2023), we see that the magnitude profile and weight difference consistently increases as the dimensionality decreases as plotted in Fig. 5. The first two principal components (PCs) constitute an unsuitable representation of these data, and the corresponding magnitude weight difference is noticeably higher than for any higher-dimensional projections. Meanwhile, the first 50 PCs capture the trend in the magnitude function of the original space, confirming that magnitude difference accurately encodes the loss of information introduced via preprocessing. Next, we compare the visualisation of these data using the first two PCs against t-SNE, UMAP or PHATE plots each computed after an initial reduction to 30, 40, or 50 PCs, following common practice. In terms of magnitude weight difference, from the full results summarised in Table 4 in the appendix, we find that UMAP, with an average magnitude weight difference of $0.395$, consistently performs better than t-SNE, with a mean difference of $0.443$, which however achieved a lower dissimilarity than PHATE with a value of $0.456$. Meanwhile, the magnitude weight difference for two PCs

is higher than for t-SNE supporting that, compared to t-SNE, this linear projection better preserves local density and pairwise distances as given in Table 1. Interestingly, these results confirm existing analyses, which found that UMAP is often more faithful to the true biological signal than t-SNE (Heumos et al., 2023), and that it better captures the distinct cell clusters visible in the data than PHATE, which is more suitable for describing continuous cell states (Moon et al., 2019). Indeed the visualisations given in Fig. 4 support the observation that PHATE introduces visibly more global distortion than the other manifold learning methods as measured by its noticeably higher magnitude weight difference. Nevertheless, PHATE still achieves better results than UMAP in terms of local density preservation and does comparably well in terms of trustworthiness and distance correlation. Magnitude can thus successfully detect that the continuous trajectory structures shown by PHATE are an unsuitable choice for representing the clusters in these data, whereas alternative measures fail to make this distinction.

## 6 DISCUSSION

Using *magnitude*, a multi-scale invariant of metric spaces, we have derived a novel measure of the quality of embedding algorithms. To this end, we have formalised innovative notions *difference* between appropriately-scaled magnitude functions as well as magnitude weights, and demonstrated their utility as embedding quality measures across multiple representation learning tasks, using both real and simulated data. Magnitude reliably detects the underlying ground truth in the context of manifold learning and turns out to be capable of determining the most faithful visualisation of single-cell data, thus preserving both the local and the global geometry of a dataset. For future work, we believe that the proposed magnitude profile differences exhibit a strong potential for applications to unaligned spaces, for example when integrating multimodal single-cell data. There are also promising opportunities to further advance the scalability of magnitude calculations, leveraging either subsampling or numerical approximation techniques. Finally, integrating magnitude into deep learning models would be beneficial to obtain novel geometry-based regularisation strategies.

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

# A APPENDIX

## A.1 PROOFS

**Lemma 4.1** (Isometry invariance). *Given two isometric spaces $X, Y$, we have $\mathrm{Mag}_X = \mathrm{Mag}_Y$.*

*Proof.* Let $(X, d_X)$ and $(Y, d_Y)$ be metric spaces with cardinality $n$ and let $f \colon X \to Y$ denote their isometry. Then, the similarity matrix of $X$ is $\zeta_{ij}^X = e^{-d_X(a_i, a_j)}$. Since $f$ is an isometry, we have, $d_X(a_i, a_j) = d_Y(f(a_i), f(a_j))$. Hence, $\zeta_{ij}^X = e^{-d_X(a_i, a_j)} = e^{-d_Y(f(a_i), f(a_j))} = \zeta_{ij}^Y$. Since $X$ and $Y$ have the same similarity matrix, we have $\mathrm{Mag}_X = \mathrm{Mag}_Y$. $\qquad\square$

**Lemma 4.2.** *Let $\|A\|_2 := \sup\{\|Ax\|_2 : x \in \mathbb{R}^n \text{ with } \|x\|_2 = 1\}$ refer to the* induced 2-norm *for matrices, and let $A, B$ be two $n \times n$ matrices with $\|A - B\|_2 \leq \epsilon$. Moreover, let $f(M) := \mathbb{1}^\top M \mathbb{1}$. Then $\|f(A) - f(B)\|_2 \leq n\epsilon$.*

*Proof.* Because $\|\cdot\|_2$ is a *consistent* norm, we have $\|f(M)\|_2 \leq \|\mathbb{1}^\top\|_2 \|M\|_2 \|\mathbb{1}\|_2 = n\|M\|_2$ for all $n \times n$ matrices $M$. Without loss of generality, assume that $\|f(A)\|_2 \geq \|f(B)\|_2$ and $\|A\|_2 \geq \|B\|_2$. Thus, $\|f(A)\|_2 - \|f(B)\|_2 \leq d(\|A\|_2 - \|B\|_2) \leq d(\|A - B\|_2) = n\epsilon$. $\qquad\square$

**Lemma 4.3.** *Let $(X, d_X)$ and $(Y, d_Y)$ be two metric spaces with corresponding distance matrices $D_X, D_Y$ and cardinality $n$. For all $\epsilon > 0$, there exists $\delta > 0$ such that if $|D_X - D_Y| < \delta$ holds elementwise, then $\|\zeta_X - \zeta_Y\|_2 \leq \epsilon$.*

*Proof.* As a consequence of the continuity of the exponential function, we know that there is $\delta$ such that $|\zeta_X - \zeta_Y| < n^{-1}\epsilon$. The row sums of $\zeta_X - \zeta_Y$ are therefore upper-bounded by $\epsilon$. We thus have $\|\zeta_X - \zeta_Y\|_2 \leq \epsilon$ (Minc, 1988, Theorem 1.1, p. 24). $\qquad\square$

## A.2 METHODS FOR COMPUTING MAGNITUDE

When computing magnitude differences in practice, only two parameter choices are necessary - the choice of comparable convergence points and the choice of evaluation values at which to empirically evaluate magnitude. For the Swiss Roll example, we choose to scale all magnitude functions until 95% of points are visible and evaluate each function across 64 equally spaced intervals. For the MNIST and the PBMCs data, we again compute the magnitude difference between lower dimensional representations and the original datasets until a desired value of magnitude chosen to equal 95% of the cardinality across 32 intervals. Across all examples, we approximate the difference in magnitude functions and difference in magnitude weight functions using Romberg integration. Any duplicate observations in the datasets or embeddings analysed throughout this study were removed to guarantee that magnitude can be computed for these spaces.

## A.3 SINGLE CELL DATA ANALYSIS

We first try out method on a well-known dataset on human peripheral blood mononuclear cells (PBMCs) from one healthy donor provided by 10x Genomics that is available through scanpy and has been processed following Seurat's guided tutorial (Satija et al., 2015; Wolf et al., 2018). After processing, the data consists of 2,638 cells with 1,838 highly variable genes and eight annotated cell type clusters. As given in Figure 5, we first compare the magnitude difference between the "original" space, that is the space spanned by all genes, to the first 2, 4, 5, 10, 30, 40 and 50 principal components calculated from these data. Indeed, we see that as the number of PCs decreases, the magnitude difference consistently increases. The two dimensional representation is by far the worse projection, whereas the 50 PCs closest capture the trend in magnitude of the original space confirming our intuition.

## A.4 EXTENDED RESULTS ON MAGNITUDE FUNCTION AND WEIGHT DIFFERENCES

For completeness, Figure 6 and 7 report on specific examples of the re-scaled magnitude functions and weight differences computed across multiple scales for the MNIST data and Swiss Roll simulation considered in the main body of the paper.

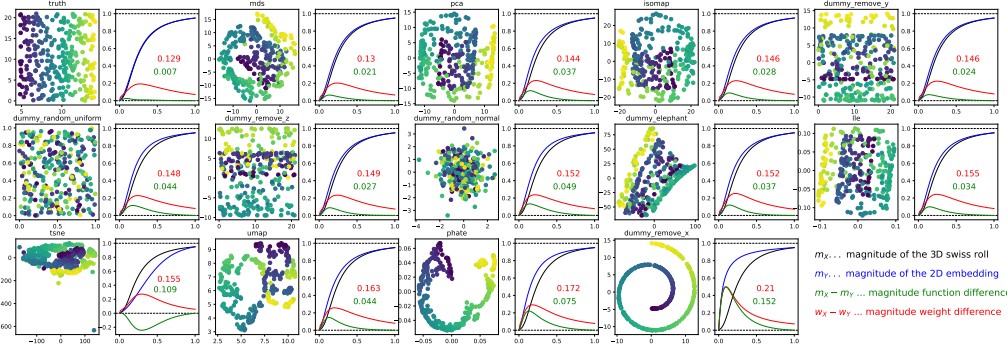

Figure 6: **Magnitude functions and magnitude differences amongst different embeddings for the Swiss Roll.** Magnitude reliably detects the ground truth.

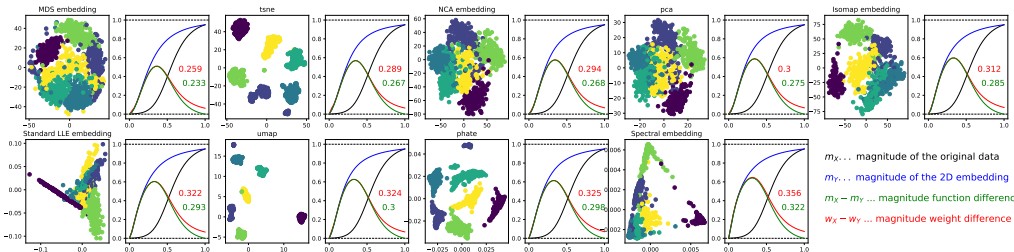

Figure 7: **Magnitude functions and magnitude differences amongst different embeddings for the digits data.** Smaller magnitude differences are related to better clustering preservation.

### A.5 EXTENDED COMPARISON BETWEEN EMBEDDING QUALITY MEASURES

In the results reported throughout this paper, we choose to compare the magnitude weight difference to the Spearman correlation in pairwise Euclidean distances (DC), the Spearman correlation in pairwise spectral distances (DCS), the preservation of local density (DP) and the RMSE in Euclidean distances. Further, the tables below shows a more complete comparison between other embedding quality measures, dimensionality reduction methods and both magnitude weight and magnitude profile differences extending the results in the main texts. For all neighbourhood-based embedding quality measures k=30 was choosen by default.

Finally, we choose to report on the embedding quality measures listed below across an extended list of dimensionality reduction methods for each of the datasets considered throughout this study as reported in Table 2, Table 3 and Table 4:

- MW_2: Magnitude weight difference using Euclidean distances
- MW_2: Magnitude profile difference using Euclidean distances
- DC: Spearman correlation coefficient between pairwise Euclidean distances
- DCS: Spearman correlation coefficient between pairwise spectral distances
- RMSE: Root mean squared error between pairwise Euclidean distances
- stress: Normalized stress measure
- DP: Local density preservation
- TW: Trustworthiness
- NL: Neighbourhood loss
- C: Continuity
- M: Mean relative rank error between original and embedding
- M1: Mean relative rank error between embedding and original
- DG: Density preservation global (Gaussian kernel)
- DKLG: Kullback–Leibler divergence between the density estimates (Gaussian kernel)
- DGL: Density preservation global (Laplacian kernel)
- DKLGL: Kullback–Leibler divergence between the density estimates (Laplacian kernel)
- SS: Silhouette score

## A.6   EXAMPLE OF CALCULATING MAGNITUDE FOR PLANETS DATA

We further decided to add another toy example on a dataset with measurements on eight planets to demonstrate the calculation of magnitude weights and the magnitude function.

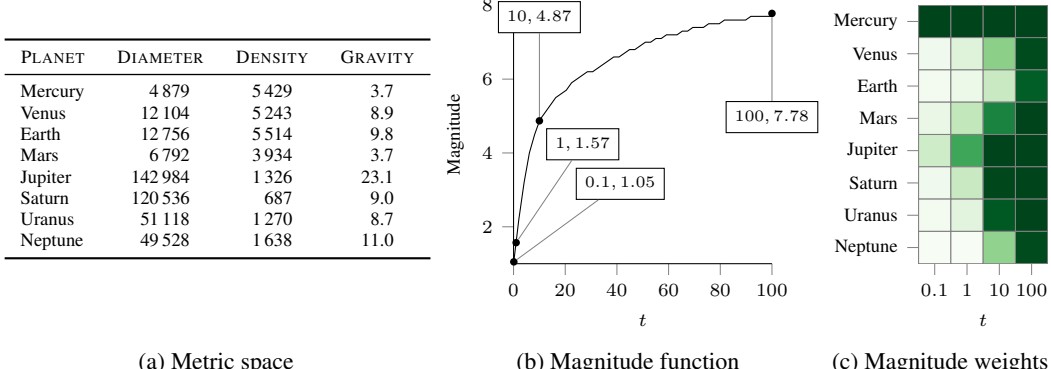

| PLANET | DIAMETER | DENSITY | GRAVITY |
|--------|----------|---------|---------|
| Mercury | 4 879 | 5 429 | 3.7 |
| Venus | 12 104 | 5 243 | 8.9 |
| Earth | 12 756 | 5 514 | 9.8 |
| Mars | 6 792 | 3 934 | 3.7 |
| Jupiter | 142 984 | 1 326 | 23.1 |
| Saturn | 120 536 | 687 | 9.0 |
| Uranus | 51 118 | 1 270 | 8.7 |
| Neptune | 49 528 | 1 638 | 11.0 |

(a) Metric space          (b) Magnitude function          (c) Magnitude weights

Figure 8: **Example of magnitude** for a metric space with eight points.

**Example A.1.** *Consider the magnitude function of the metric space consisting of 5 data points $(X, d)$ where each observation corresponds to standard-scaled measurements of the mass, diameter and density of one of the eight planets in our solar system as plotted in Figure 8. When the zooming scale $t$ is very small, for example $t = 0.01$, the magnitude weights of all points sum up to 1 and the space effectively looks like one point. Following this, magnitude grows so that at $t = 1$, around $1.6$ distinct points are visible, which increases to almost five at scale $t = 10$. Finally, when $t$ is large, for example $t = 100$, all the points are clearly separated, their magnitude weights converge to one and the value of magnitude approaches 8, which equals the cardinality of the space.*

Table 2: Quality measures of different embeddings for the Swiss Roll.

| | Emb | MW_2 | MF_2 | DC | DCS | RMSE | stess | DP | TW | NL | C | M | M1 | DG | DKLG | DGL | DKLGL |
|---|---|---|---|---|---|---|---|---|---|---|---|---|---|---|---|---|---|
| 0 | original | 0.0 | 0.0 | 1.0 | 0.026 | 0.0 | 0.0 | 1.0 | 1.0 | 0.0 | 1.0 | 0.0 | 0.0 | 0.0 | 0.0 | 0.0 | 0.0 |
| 1 | truth | 0.129 | 0.007 | 0.459 | 0.081 | 8.59 | 0.894 | 0.409 | 0.797 | 0.589 | 0.868 | 0.109 | 0.079 | 0.222 | 0.034 | 0.237 | 0.041 |
| 2 | mds | 0.13 | 0.021 | 0.883 | -0.001 | 3.214 | 0.202 | 0.52 | 0.928 | 0.347 | 0.966 | 0.077 | 0.037 | 0.084 | 0.007 | 0.127 | 0.014 |
| 3 | pca | 0.144 | 0.037 | 0.861 | -0.029 | 3.89 | 0.274 | 0.343 | 0.901 | 0.367 | 0.973 | 0.109 | 0.027 | 0.156 | 0.021 | 0.195 | 0.034 |
| 4 | isomap | 0.146 | 0.028 | 0.844 | -0.032 | 8.661 | 0.366 | 0.248 | 0.923 | 0.343 | 0.972 | 0.087 | 0.029 | 0.177 | 0.027 | 0.23 | 0.045 |
| 5 | d_no_y | 0.146 | 0.024 | 0.799 | 0.029 | 4.697 | 0.349 | 0.466 | 0.869 | 0.452 | 0.963 | 0.148 | 0.035 | 0.173 | 0.019 | 0.189 | 0.025 |
| 6 | ran_un_9 | 0.148 | 0.044 | -0.001 | 0.001 | 15.788 | 27.714 | 0.028 | 0.522 | 0.904 | 0.52 | 0.493 | 0.495 | 0.293 | 0.065 | 0.298 | 0.07 |
| 7 | d_no_z | 0.149 | 0.027 | 0.709 | 0.058 | 5.701 | 0.45 | 0.541 | 0.825 | 0.499 | 0.952 | 0.203 | 0.043 | 0.151 | 0.017 | 0.154 | 0.021 |
| 8 | d_r_n | 0.152 | 0.049 | -0.008 | -0.027 | 14.631 | 7.215 | -0.073 | 0.523 | 0.896 | 0.523 | 0.491 | 0.49 | 0.284 | 0.073 | 0.368 | 0.137 |
| 9 | d_el | 0.152 | 0.037 | 0.713 | 0.025 | 68.847 | 0.819 | 0.272 | 0.862 | 0.511 | 0.933 | 0.159 | 0.055 | 0.171 | 0.019 | 0.206 | 0.032 |
| 10 | lle | 0.155 | 0.034 | 0.852 | -0.006 | 16.153 | 139.889 | 0.449 | 0.897 | 0.396 | 0.971 | 0.121 | 0.03 | 0.148 | 0.018 | 0.171 | 0.026 |
| 11 | tsne | 0.155 | 0.109 | 0.706 | -0.018 | 142.14 | 0.905 | 0.433 | 0.943 | 0.328 | 0.913 | 0.045 | 0.077 | 0.173 | 0.031 | 0.173 | 0.031 |
| 12 | umap | 0.163 | 0.044 | 0.643 | 0.035 | 12.411 | 2.947 | 0.408 | 0.981 | 0.259 | 0.946 | 0.019 | 0.033 | 0.207 | 0.034 | 0.2 | 0.031 |
| 13 | phate | 0.172 | 0.075 | 0.406 | -0.001 | 16.193 | 201.335 | 0.316 | 0.91 | 0.41 | 0.846 | 0.066 | 0.071 | 0.198 | 0.03 | 0.205 | 0.034 |
| 14 | d_no_x | 0.21 | 0.152 | 0.844 | -0.01 | 4.359 | 0.319 | 0.426 | 0.89 | 0.445 | 0.966 | 0.137 | 0.04 | 0.134 | 0.013 | 0.135 | 0.015 |

Table 3: Quality measures of different embeddings for MNIST.

| | Emb | MW2 | MF2 | DC | DCS | RMSE | stess | DP | TW | NL | C | M | M1 | RIE | RIA | DG | DKLG | DGL | DKLGL | SS |
|---|---|---|---|---|---|---|---|---|---|---|---|---|---|---|---|---|---|---|---|---|
| 0 | Or | 0.0 | 0.0 | 1.0 | 0.074 | 0.0 | 0.0 | 1.0 | 1.0 | 0.0 | 1.0 | 0.0 | 0.0 | 1.0 | 1.0 | 0.0 | 0.0 | 0.0 | 0.0 | 0.205 |
| 1 | MDS | 0.259 | 0.233 | 0.779 | 0.064 | 15.314 | 0.327 | 0.583 | 0.907 | 0.646 | 0.93 | 0.098 | 0.069 | 0.922 | 0.854 | 0.238 | 0.042 | 0.189 | 0.029 | 0.297 |
| 2 | tsne | 0.289 | 0.267 | 0.571 | -0.015 | 21.343 | 0.374 | 0.425 | 0.985 | 0.356 | 0.98 | 0.012 | 0.016 | 0.918 | 0.725 | 0.314 | 0.085 | 0.25 | 0.05 | 0.673 |
| 3 | NCA | 0.294 | 0.268 | 0.687 | 0.049 | 28.25 | 0.409 | 0.1 | 0.883 | 0.674 | 0.953 | 0.118 | 0.044 | 0.923 | 0.74 | 0.366 | 0.102 | 0.351 | 0.097 | 0.374 |
| 4 | pca | 0.3 | 0.275 | 0.693 | 0.047 | 23.821 | 0.808 | 0.175 | 0.875 | 0.677 | 0.95 | 0.133 | 0.046 | 0.917 | 0.854 | 0.393 | 0.118 | 0.373 | 0.106 | 0.326 |
| 5 | Iso | 0.312 | 0.285 | 0.649 | 0.035 | 41.245 | 0.494 | 0.484 | 0.902 | 0.613 | 0.968 | 0.106 | 0.033 | 0.922 | 0.803 | 0.366 | 0.109 | 0.298 | 0.069 | 0.439 |
| 6 | SLLE | 0.322 | 0.293 | 0.416 | 0.075 | 49.265 | 810 | 0.144 | 0.825 | 0.757 | 0.883 | 0.174 | 0.095 | 0.708 | 0.033 | 0.364 | 0.129 | 0.514 | 0.24 | 0.084 |
| 7 | umap | 0.324 | 0.3 | 0.521 | -0.003 | 37.667 | 2.803 | 0.127 | 0.985 | 0.369 | 0.978 | 0.018 | 0.018 | 0.855 | 0.632 | 0.379 | 0.143 | 0.28 | 0.065 | 0.736 |
| 8 | phate | 0.325 | 0.298 | 0.578 | 0.011 | 49.279 | 1190 | 0.388 | 0.966 | 0.49 | 0.974 | 0.036 | 0.022 | 0.891 | 0.642 | 0.329 | 0.086 | 0.263 | 0.055 | 0.572 |
| 9 | Sp | 0.356 | 0.322 | 0.554 | -0.001 | 49.312 | 8516 | 0.43 | 0.923 | 0.628 | 0.96 | 0.084 | 0.043 | 0.815 | 0.713 | 0.456 | 0.193 | 0.431 | 0.136 | 0.55 |

Table 4: Quality measures of different embeddings for the single cell data.

| | Em | MW1 | MF1 | MW2 | MF2 | DC | DCS | RMSE | stess | DP | TW | NL | C | M | M1 | RIE | RIA | DG | DKLG | DGL | DKLGL | SS |
|---|---|---|---|---|---|---|---|---|---|---|---|---|---|---|---|---|---|---|---|---|---|---|
| 0 | Or | 0.0 | 0.0 | 0.0 | 0.0 | 1.0 | -0.024 | 0.0 | 0.0 | 1.0 | 1.0 | 0.0 | 1.0 | 0.0 | 0.0 | 1.0 | 1.0 | 0.0 | 0.0 | 0.0 | 0.0 | 0.01 |
| 1 | 50 | 0.193 | 0.019 | 0.178 | 0.078 | 0.758 | 0.067 | 37.5 | 1.903 | 0.879 | 0.84 | 0.887 | 0.869 | 0.149 | 0.113 | 0.999 | 0.991 | 0.149 | 0.017 | 0.118 | 0.012 | 0.128 |
| 2 | 40 | 0.198 | 0.031 | 0.186 | 0.089 | 0.739 | 0.072 | 38.5 | 2.051 | 0.873 | 0.822 | 0.9 | 0.855 | 0.168 | 0.125 | 0.999 | 0.989 | 0.153 | 0.018 | 0.125 | 0.013 | 0.146 |
| 3 | 30 | 0.203 | 0.049 | 0.196 | 0.106 | 0.71 | 0.078 | 39.8 | 2.264 | 0.861 | 0.795 | 0.919 | 0.833 | 0.197 | 0.147 | 0.999 | 0.986 | 0.159 | 0.02 | 0.134 | 0.015 | 0.176 |
| 4 | 20 | 0.212 | 0.077 | 0.214 | 0.131 | 0.654 | 0.084 | 41.7 | 2.594 | 0.821 | 0.747 | 0.945 | 0.795 | 0.251 | 0.186 | 0.999 | 0.975 | 0.169 | 0.022 | 0.154 | 0.02 | 0.233 |
| 5 | 10 | 0.233 | 0.135 | 0.247 | 0.186 | 0.583 | 0.09 | 43.9 | 3.089 | 0.721 | 0.674 | 0.969 | 0.738 | 0.328 | 0.248 | 0.999 | 0.96 | 0.179 | 0.025 | 0.185 | 0.028 | 0.362 |
| 6 | 5 | 0.276 | 0.216 | 0.303 | 0.266 | 0.529 | 0.096 | 45.6 | 3.512 | 0.583 | 0.635 | 0.977 | 0.718 | 0.365 | 0.268 | 0.996 | 0.93 | 0.186 | 0.027 | 0.216 | 0.037 | 0.458 |
| 7 | 4 | 0.291 | 0.244 | 0.32 | 0.293 | 0.519 | 0.106 | 46.1 | 3.657 | 0.568 | 0.626 | 0.977 | 0.709 | 0.375 | 0.274 | 0.987 | 0.934 | 0.188 | 0.027 | 0.2 | 0.033 | 0.42 |
| 8 | 40_u | 0.35 | 0.33 | 0.39 | 0.379 | 0.331 | 0.043 | 49.8 | 6.357 | 0.193 | 0.625 | 0.974 | 0.675 | 0.377 | 0.326 | 0.257 | 0.193 | 0.262 | 0.056 | 0.27 | 0.056 | 0.492 |
| 9 | 30_u | 0.354 | 0.335 | 0.397 | 0.386 | 0.352 | 0.054 | 49.3 | 5.801 | 0.173 | 0.628 | 0.975 | 0.712 | 0.375 | 0.277 | 0.182 | -0.502 | 0.277 | 0.063 | 0.277 | 0.06 | 0.496 |
| 10 | 50_u | 0.36 | 0.339 | 0.399 | 0.388 | 0.348 | 0.053 | 50.3 | 7.027 | 0.174 | 0.626 | 0.974 | 0.719 | 0.373 | 0.262 | 0.434 | 0.07 | 0.265 | 0.057 | 0.265 | 0.055 | 0.492 |
| 11 | 2 | 0.397 | 0.375 | 0.437 | 0.424 | 0.447 | 0.07 | 48.1 | 4.767 | 0.407 | 0.611 | 0.98 | 0.716 | 0.394 | 0.268 | 0.255 | 0.382 | 0.225 | 0.046 | 0.328 | 0.086 | 0.408 |
| 12 | 50_t | 0.395 | 0.376 | 0.438 | 0.427 | 0.334 | 0.065 | 29.3 | 0.781 | -0.016 | 0.643 | 0.969 | 0.735 | 0.34 | 0.248 | 0.173 | -0.413 | 0.245 | 0.052 | 0.208 | 0.034 | 0.453 |
| 13 | 40_t | 0.403 | 0.381 | 0.442 | 0.43 | 0.304 | 0.104 | 28.9 | 0.803 | -0.233 | 0.635 | 0.973 | 0.602 | 0.36 | 0.388 | 0.4 | -0.327 | 0.234 | 0.044 | 0.203 | 0.033 | 0.4 |
| 14 | 30_t | 0.407 | 0.387 | 0.449 | 0.438 | 0.371 | 0.068 | 26.3 | 0.523 | -0.041 | 0.643 | 0.971 | 0.727 | 0.341 | 0.257 | 0.179 | -0.562 | 0.229 | 0.045 | 0.204 | 0.033 | 0.431 |
| 15 | 30_p | 0.415 | 0.389 | 0.453 | 0.439 | 0.348 | 0.049 | 56 | 1786 | 0.271 | 0.62 | 0.978 | 0.702 | 0.377 | 0.292 | 0.295 | 0.814 | 0.331 | 0.09 | 0.435 | 0.156 | 0.598 |
| 16 | 40_p | 0.415 | 0.39 | 0.454 | 0.44 | 0.334 | 0.044 | 56 | 1817 | 0.322 | 0.618 | 0.978 | 0.691 | 0.384 | 0.296 | 0.277 | 0.856 | 0.323 | 0.086 | 0.428 | 0.15 | 0.617 |
| 17 | 50_p | 0.421 | 0.395 | 0.46 | 0.444 | 0.358 | 0.045 | 56 | 1821 | 0.298 | 0.622 | 0.976 | 0.714 | 0.379 | 0.273 | 0.421 | 0.558 | 0.327 | 0.088 | 0.435 | 0.156 | 0.62 |

