# OpenReview forum: "Metric Space Magnitude for Evaluating Unsupervised Representation Learning"
_ICLR.cc/2024/Conference — Submitted to ICLR 2024_

### Official Review · Reviewer_WDGw · 2023-10-28

**Soundness:** 2 fair
**Presentation:** 4 excellent
**Contribution:** 3 good
**Rating:** 5
**Confidence:** 3

**Summary:**

The paper introduces a concept called "magnitude" for metric spaces, serving as an invariant to measure the effective size of a space across various scales. This magnitude considers both geometric and topological aspects, making it useful for unsupervised representation learning tasks. The authors propose a dissimilarity measure between magnitude functions of finite metric spaces, offering a stable quality assessment for dimensionality reduction tasks. The measure proves stability under data perturbations, is computationally efficient, and facilitates a thorough multi-scale comparison of embeddings. Experimental results validate the measure's utility. The paper establishes a scale-invariant framework for quantifying the difference between magnitudes, links this difference to embedding quality, and demonstrates empirical effectiveness on simulated and real data.

**Strengths:**

The paper's strengths lie in its:

- Clarity and Organization: The clarity of writing and well-organized structure contribute to an easy and accessible read, enhancing the paper's overall effectiveness.

- Accessible Background: The inclusion of a background that caters to readers outside the specialized field ensures that a broader audience can understand and engage with the content. This accessibility promotes the dissemination of the concepts introduced.

- Comprehensive Approach: By addressing both theoretical and computational aspects of the proposed magnitude, the paper offers a well-rounded exploration of the concept. This comprehensive coverage enhances the paper's credibility and applicability in both theoretical and practical contexts.

**Weaknesses:**

The weaknesses of the paper include:

- Limited Convincing Power in Experimental Results: The paper's diverse range of experiments is noted, but the absence of statistical uncertainty in the reported results diminishes their convincing power. Without incorporating measures of uncertainty, readers may question the generalizability of the experimental findings to the broader population and the long-term effectiveness of the proposed method.

- Lack of Statistical Analysis: The paper seems to lack a robust statistical analysis of the experimental results. Statistical analyses, such as confidence intervals or hypothesis testing, are crucial for establishing the reliability and significance of the observed effects. Without this, the strength of the experimental evidence may be undermined.

Addressing these weaknesses through the inclusion of statistical uncertainty measures and a more rigorous statistical analysis would strengthen the paper's overall credibility and enhance the confidence readers have in the validity of the experimental results.

**Questions:**

Please see above

---

> ### Author Response · Authors · 2023-11-20
>
> **Statistical analysis:**
>
> • We agree that it would be desirable to add secondary statistical analysis to the main results reported in the paper. However, there are some open problems with regards to this analysis.
>
> • Confidence intervals would be an obvious choice for estimating the statistical uncertainty of the reported magnitude differences. However, the true distribution of magnitude and magnitude differences is not known, prohibiting the use of parametric methods.
>
> • In terms of non-parametric statistics, variations of bootstrapping methods are most commonly used for estimating confidence intervals from empirical distributions. As a core idea, bootstrapping relies on random sampling with replacement. Unfortunately, metric space magnitude does not allow for this sampling strategy as it cannot be computed from data with duplicate points (as then the similarity matrix will not be invertible and magnitude will not exist).
>
> • Estimating uncertainty via subsampling could be an alternative, but is considerably less statistically sound as resamples would not be independent and identically distributed. Further, whether magnitude could be approximated via subsampling has not yet been investigated.
>
> • Given these limitations, estimating the statistical uncertainty of magnitude remains a challenging task and will be the topic of further work.
>
> • We further refrain from reporting p-values for the magnitude differences. This is motivated by well-known criticisms on the overuse of p-values and the fact that reporting statistical significance for embedding quality measures would require a suitable null model. In practice however, the ideal $d \ll D$ dimensional embedding is generally unknown and it is not at all obvious how to fulfill the null hypothesis.

---

> > ### Comment · Reviewer_WDGw · 2023-11-23
> >
> > I appreciate your attention to my concerns, and I've chosen to stick with my current score.

---

### Official Review · Reviewer_t2Dn · 2023-10-28

**Soundness:** 2 fair
**Presentation:** 2 fair
**Contribution:** 2 fair
**Rating:** 5
**Confidence:** 4

**Summary:**

This paper focuses on the metric space magnitude which is defined as a measure of the similarity matrix of a set of instances. The authors proposed two new definitions of magnitude differences (namely dissimilarity). They provided theoretical analysis to validate the robustness and invariant properties of their proposed metrics. Numerical experiments on some classical unsupervised representation learning approaches demonstrate the soundness of the proposed method.

**Strengths:**

1). The topic studied in this paper is interesting and I really appreciate the authors' efforts on the theoretical formulations and analysis results.

2). Experiments on classical manifold/unsupervised approaches with real-world data validate that the proposed magnitude difference is indeed a reasonable metric to evaluate the consistency between two metric spaces.

**Weaknesses:**

1). The novelty is somewhat limited. Considering that the metric space magnitude is already an existing work and has been widely applied in mathematics and machine learning, the authors's contribution is actually providing the difference between the magnitudes of two metric spaces.

2). The technical difficulty is trivial. When I check the theoretical results in lemmas and corollaries, it seems that most of them are very straightforward conclusions from the corresponding definitions (the proofs are actually very short as shown in the appendix).

3). The motivation and experiments are not very solid to me. The effectiveness of unsupervised dimensionality reduction can already be evaluated by their original loss functions such as distance preservation and reconstruction error. Why do we still need the magnitude difference to validate their effectiveness? Second, how such a new space metric help us in our algorithm desgin? For example, can we integrate this new magnitude difference into exsiting unsupervised learning to obtain some better dimensionality reduction results? I was looking forward to seeing these results but I failed to find any of them in the experiment section.

**Questions:**

See the weaknesses.

---

> ### Author Response · Authors · 2023-11-20
>
> **Novelty of magnitude:**
>
> • It is worth noting that metric space magnitude has not been widely applied in neither mathematics nor machine learning. In the broadest context, existing applications of metric space magnitude are limited to a handful of papers tackling specific data analysis tasks such as biodiversity estimation (Meckes 2015), boundary detection (Bunch 2020), outlier detection (Bunch 2020), convex hull approximation (Fung 2019), edge detection in images (Adamer 2021) or neural network learning theory (Andreeva 2023).
>
> • Although magnitude weights and functions are known in the mathematical literature, they have received little attention from both computational mathematicians and machine learning practitioners. While alternative geometric methods and multiscale summaries, such as persistent homology, are relatively well established tools appearing in hundreds of publications, applications of magnitude are considerably less explored.
>
> • Even in terms of theoretical work, key results that would enable successful applications of metric space magnitude to real data do not yet exist. In particular, proofs on the stability of magnitude, its statistical properties or efficient approximation methods have not been developed.
>
> ---
>
> **Novelty of our contributions:**
>
> • We aim to address the aforementioned gaps in existing research by being the first to propose a distance between the magnitude weights and functions of different metric spaces and applying these for evaluating embedding quality, which is the first application of magnitude to unsupervised representation learning. However, these are not our only contributions.
>
> • The novelty in our work also lies in speeding up magnitude computation, making it more accessible for larger datasets. This adresses the main bottleneck for wider adoption of magnitude in practice, which has been the computational complexity of inverting a matrix.
>
> • In terms of technical difficulty, we agree that there is room for more theoretical advancements. Some straightforward corollaries in the paper (e.g. corollary 4.1 and 4.2) have been chosen to illustrate the impact of re-scaling a space and make the material more accessible and focus on motivating the proposed indicators.
>
> • We would also like to note that the stability result introduced in Lemma 4.2 and 4.3 is novel. Even in theoretical literature, although showing robustness to noise is vital for data analysis tasks, no known stability guarantee exists. Indeed, proving the stability of magnitude is not a trivial conclusion.
>
>
>
> ---
>
> **Motivation and experiments:**
>
> • While algorithms’ original loss functions can also be used to evaluate their performance, these measures will of course be biased towards the models trained on them and do not always constitute suitable choices for secondary evaluation. The use of embedding quality measures extends beyond these as a guide to inform about the preservations of properties not explicitly involved in model training.
>
> • Indeed, we have demonstrated cases where magnitude difference is superior to alternative embedding quality measures by capturing geometric information beyond, for example, distance or density preservation.
>
> • In the case of visualization, for example, magnitude is aware of “zooming” and tracking the “effective number of points” corresponding to an intuitive understanding of what makes a good picture. Indeed, the experiments show that magnitude can help choose better visualizations. A particularly relevant application for visualisation is in single cell data analysis, where the use of embedding quality measures is important for understanding the amount of both local and global distortion introduced by dimensionality reduction.
>
>
>
> ---
>
> **Integrating magnitude into unsupervised learning:**
>
> • We are excited about the reviewer's interest in further applications of magnitude to unsupervised representation learning and the integration of magnitude into model training itself.
>
> • Incorporating magnitude into the model’s loss is not a straightforward task and requires more theoretical advancements, which will be the subject of future work. However, this paper is meant as a first exploration of metric space magnitude as an indicator of embedding quality to first overcome some of the practical hurdles on computing magnitude on real data, and second, to present an investigation on why smaller magnitude differences correspond to better representations.

---

### Official Review · Reviewer_545Y · 2023-11-07

**Soundness:** 2 fair
**Presentation:** 2 fair
**Contribution:** 2 fair
**Rating:** 3
**Confidence:** 2

**Summary:**

This paper proposes a method using magnitude in metric space as an evaluation method for dimensionality reduction methods. Specifically, the paper defines two indicators and compares them with traditional indicators on multiple data sets.

**Strengths:**

The concept of Magnitude on the metric space is extended to define a suitable metric for evaluating dimensionality reduction.

**Weaknesses:**

The first point of concern is that I do not know what the paper is trying to argue. If the claim is that the proposed method is an indicator that can determine whether it is a good dimensionality reduction or not, it is necessary to clarify what makes it a good dimensionality reduction. If this were global isometry (which seems to be the case from the theorem that follows), then the experiments should be compared in this light, but this does not seem to be the case. The experiment seems to be just an observation, and I am not sure if the proposed method is a good one.

The second concern is in terms of novelty: the magnitude function is already known and the difference defined is the simplest one. It should be clarified whether the introduction of Magnitude into the evaluation of embedding performance is novel or whether the construction of a specific indicator is novel. "Our contribution" section seems to indicate that the evaluation of the experiment and the proof of the theorem are additional contributions. Whichever points are novel, it should be clear why they are necessary. The motivation for introduction is written, but it is similar to the first point that it should be clear why the proposed method is superior.

Third, it is unfriendly to readers in several respects.For example,
- The main proposal in this paper appears to be Def. 4.2 and 4.3, but it is written in a way that makes it unclear that it is a proposed method. Fig. 2 helped me recognize this point, but it would not be sufficient as a document. (Is it a typo that in Fig.2 it is Magnitude function difference, but in Def.4.2 it is Magnitude profile difference?)
- It seems to me that Def4.3 must assume that the X and Y components are aligned correspondingly, but it does not say that. (Otherwise, they are not unique according to the order.) There is a reference at the beginning of Section 4, but it does not specify that the alignment condition is a prerequisite.

**Questions:**

- Regarding the definition of re-scaled in Def.4.1, it is multiplied by $t_{conv}$, but seems to emphasize scale more. Is $s/t_{conv}$ incorrect?
- Scale-invariant is claimed as an advantage of the proposed method, which I believe means that it does not require the setting of hyperparameters in the neighborhood. On the other hand, magnitude measures global features when the scale parameter is small and local features when it is large. In the end, the proposed index is considered to be the sum (average) of the differences in features calculated by varying the hyperparameters by some rule. From this perspective, even with the conventional method that requires hyperparameter settings, the objective can be achieved by changing the hyperparameters and taking the sum (average) of the calculated values. Is there any reason why scale-invariant can be achieved because of Magnitude?
- From a representation learning perspective, affine transformations, i.e., enlargement and reduction, are not so essential differences. From this perspective, it would be more valuable for the proposed method to be an indicator that the expansion and contraction can be automatically ignored by re-scaling.

---

> ### Author Response · Authors · 2023-11-20
>
> **Main claim:**
>
> • The main claim of our work is indeed that a good representation is one that better resembles the geometric structure of the original higher dimensional space by faithfully preserving magnitude in lower dimensions and achieving smaller magnitude differences.
>
> • Motivated by this, the use of magnitude to indicate embedding quality is not limited solely to isometric embeddings. Indeed, we see magnitude as a tool for describing the preservation of geometric structure more generally.
>
> • While it is hard to uniformly define what makes a good representation for any application, different embedding quality measures exist with different purposes after all - magnitude difference offers a novel take on the importance of geometric information for embedding quality.
>
> • It is superior to alternative quality measures when preserving both the local and global structure of the data is important (Swiss Roll), when preserving the effective number of clusters across scales is relevant (MNIST/PBMC) or for the purpose of assessing lower dimensional visualizations.
>
> ---
>
> **Novelty:**
>
> • This work is the first to:
> * apply magnitude to the task unsupervised dimensionality reduction,
> * define differences between magnitude weights and functions of two different metric spaces,
> * link these differences to embedding quality,
> * evaluate the performance of these indicators experimentally,
> * improve the computational efficiency of calculating magnitude weights and functions using Cholesky inversion,
> * automate magnitude computation for any given dataset by automating the choice of scale parameters,
> * prove the theoretical stability of magnitude,
> * and evaluate the empirical robustness of the proposed magnitude difference experimentally.
>
> • Indeed, both the application of magnitude to unsupervised dimensionality reduction as well as the definition of a difference between magnitude functions/weights and the proposed framework to systematically compare magnitude across different metric spaces are novel. Altogether the experimental and theoretical work presented in Sections 4 and 5 are new results. In this light, we will clarify our contribution in the introduction.
>
>
> ---
>
> **Readability:**
>
> • We will clarify the definition of the proposed difference between re-scaled magnitude functions, which we term magnitude profile difference and clarify this both in Def 4.2 and Figure 2.
>
> • While both Def 4.2. and Def 4.3. assume that magnitude has been evaluated across the same scaling factors, we will highlight that the magnitude profile difference (4.2.) could also be calculated for unaligned spaces while magnitude weight difference (4.3.) requires aligned observations.
>
>
> ---
>
> **Scale-invariance:**
>
> • Scale-invariance refers to the invariance to re-scaling the distances in a metric space by a scalar as introduced in Corollary 4.2.. That is, re-scaling distances does not impact how well an embedding is perceived.
>
> • For the re-scaling introduced in Def 4.1 notice that the variable $s\in[0,1]$ is introduced and the re-parameterization is given by $Mag_X(s) = Mag_X(s*t_{conv})$ so that $Mag_X(s)\rvert_{s=1} = Mag_X(t_{conv})$. This then allows for comparing magnitude functions across comparable scaling parameters $s\in[0,1]$.
>
> • For $s \in [0, 1]$, $s/t_{conv}$ would indeed be incorrect as $Mag_X(s/t_{conv})\rvert_{s=1} \neq Mag_X(t_{conv})$.
>
>
> ---
>
> **Magnitude as a multi-scale summary:**
> • The benefit we wanted to highlight is that by numerically approximating the area between the magnitude curves, we get a holistic summary of the dissimilarity between spaces across different scales of the distances. We argue that this gives a more complete summary of the space and removes the need to manually choose a fixed scale parameter / neighborhood size.
>
> • One could also get multi-scale summaries by comparing different filtration curves - this idea is not exclusive to magnitude. It is the same concept as comparing other multi-scale summaries across multiple choices of hyperparameters (e.g. neighborhood sizes or scaling parameters).
>
> • We agree how the definition of scale-invariance vs. multi-scale summary can be confusing and will clarify this in the text.
>
>
>
> ---
>
>
> **Magnitude difference under affine transformations:**
>
> • The proposed framework has been formulated in order to be able to ignore the impact of re-scaling the distances via representation learning. As such, the proposed magnitude differences are invariant under certain affine transformations such as uniform scaling, dilation, rotation, translation and reflection.
>
> • This already allows for ignoring expansion and contraction automatically when comparing two metric spaces and this property is referred to as scale-invariance in the text.
>
> • Regarding other affine transformations, linear scaling in only one direction, if too extreme, will not be an ideal representation and not exactly preserve magnitude. Similarly, shear mapping could introduce unwanted distortion.

---

> > ### Comment · Reviewer_545Y · 2023-11-21
> >
> > Thanks to the authors for their detailed answer.
> >
> > **Main claim:**
> > I agree that it is hard to uniformly define what makes a good representation for any application. But that does not mean that it is acceptable to note it as the main claim. If we accept that, we must accept the author's self-serving indicators. In the case of this discussion, the original purpose of expressive learning should be involved, and it should be discussed that the proposed perspective is of a nature necessary to achieve the purpose.
> > The importance of an indicator to assess isometricity is understandable, since the isometricity of expressive learning is discussed in, for example, the following paper, but the discussion within the paper seems to be lacking.
> >
> > A. Nakagawa et al., QuantitativeUnderstandingofVAEasaNon-linearlyScaledIsometric Embedding, ICML2022
> >
> > **Novelty:**
> > The introduction of Magnitude into the evaluation of expressive learning is not considered novel in itself. (If we accept it, then the introduction of an outlandish tool would constitute novelty.) The novelty would be the effect of the introduction of Magnitude on expressive learning. If the expression learning is merely an example of application, then the appropriateness of this paper as an ICLR paper is questionable.
> >
> > It is difficult to recognize novelty in terms of computational complexity since there is no evaluation of them. The following paper also discusses the calculation method of Magnitude. Comparisons with the following papers may be necessary.
> >
> > E. Bunch et al., Practicalapplicationsofmetricspacemagnitudeandweighting vectors, https://arxiv.org/pdf/2006.14063.pdf
> >
> > **Scale-invariance:**
> > In this regard, I was misreading the definition.
> >
> > **Magnitude as a multi-scale summary:**
> > It seems that the authors did not understand the intent of the question. The answers sought were not given. We understand that the reason why hyperparameter selection is no longer necessary is that the features for multiple hyperparameters are acquired and merged. Since it is possible for other methods to compute features with multiple hyperparameters, it is an easy analogy to construct a method that does not require hyperparameter settings without using Magnitude. Is it an effect of the introduction of Magnitude that it can both eliminate the need for hyperparameter design and obtain a complete summary of the space?
> >
> > **Magnitude difference under affine transformations:**
> > "The proposed framework has been formulated in order to be able to ignore the impact of re-scaling the distances via representation learning. "  I think this would be true if Re-scaling was used, but I don't think it would hold otherwise. If that is my misunderstanding, please clarify the reason. My point is that re-scaling should be the main proposal. Currently, the non re-scaled one is used in the experiment, and re-scaling is additive.

---

> ### Author Response · Authors · 2023-11-23
>
> **Computational complexity:**
>
> We agree that an extensive computational benchmark is desirable and will therefore add this comparison to the manuscript.
>
> The main contribution of Bunch et. al. towards computing magnitude is the development of an inclusion / exclusion principle for magnitude. This can speed up computations in settings where $X \subset Y \subset \mathbb{R}^n$, $Mag_X$ is already known and one wants to use this knowledge to more efficiently compute $Mag_Y$.
>
> However, they do speed up the step of inverting the similarity matrix and we lack reason to believe their implementation speeds up the calculation of magnitude in general.
>
> Moreover, they do not compute the magnitude function, but the magnitude at a specified value of t, so they also do not use the full expressive power of the magnitude function and their results depend on the choice of scaling parameter.
>
>
> ---
>
>
> **Magnitude as a multi-scale summary:**
>
> For answering “Is it an effect of the introduction of Magnitude that it can both eliminate the need for hyperparameter design and obtain a complete summary of the space?”, I am afraid I might still not understand the purpose of the question. Yes, magnitude functions and weights are comprehensive summaries of a space that do not require setting fixed scale parameters or tuning hyperparameters. This is a beneficial property of magnitude functions, which are natural extensions of the single-scale version of magnitude. It is also true that there exist other summaries that fulfill the same multi-scale property, but do not use magnitude.
>
>
> ---
>
>
> **Scale-invariance and magnitude under affine transformations:**
>
> The main proposal in Def 4.2 defines a difference not between the original magnitude functions but between their re-scaled i.e. re-parametrised versions. Similarly, Def 4.3. uses the difference between the re-scaled weight vectors. We do not propose nor use a difference between the non-rescaled magnitude functions or weights in any of the experiments. In response to this feedback, we will highlight and clarify this distinction in the manuscript.
>
> We can also show the magnitude profile distance is invariant to linear re-scaling of the distances. If we take a metric space $X=(X,d)$ and its scaled version $Y=cX=(X, c \cdot d)$ for some scalar $c \in \mathbb{R}$. Say Mag_X and Mag_Y are their magnitude functions and call their re-scaled magnitude functions $M_X$ and $M_Y$. We get $\Delta Mag_{XY}= \int_S |M_X(s)-M_Y(s)| ds = \int_S |Mag_X(s \cdot x_{conv})-Mag_Y(s \cdot y_{conv})| ds = 0$
>
> ---
>
>
> **Novelty:**
>
> Studying unsupervised dimensionality reduction and the visualization or interpretation of learned representations in general, falls within the scope of representation learning and the relevant topics set out by ICLR. The introduction of magnitude to evaluate embedding quality is not merely a side though, but the main purpose of this work and the proposed indicators.
>
>
>
> ---
>
>
>
>
> **Main claim:**
>
> We choose to study representations via magnitude, because it makes sense to do so. Just like preserving other geometric properties, such as distances, density, curvature, local neighborhoods, clustering patterns or intrinsic dimensionality are obviously important, preserving magnitude is a logical extension of these aims.
>
> In particular, the goal of representation learning that was explored in our experiment is learning lower-dimensional for the task of visualisation, which is an open problem in particular when analysing single cell data. Throughout the experiments, we introduced magnitude difference as a qualitative measure for assessing 2D representations as well as used magnitude to describe the information loss introduced by projecting onto lower dimensional PCs.
>
> Given any potential gaps in our analysis, we are of course keen to try out suggestions on extended experiments that could give a more extensive picture of the expressiveness of preserving magnitude for representation learning. Nevertheless, for the purpose of preserving geometric structures important for visualisation, magnitude difference fulfills this original goal of indicating better representations across the experiments we have considered.
>
> While the isometricity of expressive learning wasn’t our main focus - we agree that it shows interesting parallels to the isometry invariance of magnitude. For our proposed magnitude differences, this invariance can easily be extended to invariance under specific types of quasi-isometries given by maps which multiply all distances by the same positive scalar i.e. $\Delta Mag_{XY}=0$ for $Y$  isometric to $cX$ for any $c \in \mathbb{R}_+$. Further, it is also of interest to study how this fits in with other notions of isometricity used in expressive learning, such as nonlinear scaled isometric embeddings.

---

> > ### Comment · Reviewer_545Y · 2023-11-23
> >
> > We thank the authors for their responses.
> >
> > The authors state that it is obviously important to preserve geometric properties, but what we are asking is why is it important? Since the original purpose of the representation study was not to retain geometric features, I wanted to show how retaining geometric features would have an effect on the original purpose. I think visualization in 2D is one example of application, but in this case it would help understanding if the geometric features are retained. However, since this is not the only important factor, it should not be used as an indicator of good or bad representation learning, but rather as a measure of the preservation of geometric features.
> >
> > Personally, I believe that preservation of geometric features is one criterion for expression learning, and this indicator may have the potential to be useful. However, this paper is not convincing enough, and as other reviewers have stated, it seems to lack novelty and experimental proof. In light of these considerations, I think it would be more useful to readers and researchers in this field if the paper is published after extensive revision and modification of its arguments, structure, and experiments, rather than being published at this point.

---

### Meta-Review · Area_Chair_NtNX · 2023-12-05

**Metareview:**

This paper proposes the concepts of magnitude profile difference and magnitude weight difference, based on the existing concept of magnitude of metric space, and use them for evaluating dimension reduction (DR) methods. The proposed method is accompanied by numerical experiments.

Reviewers generally find the proposed idea to be interesting. However, they also raise major concerns, including

- Limited novelty and technical contributions (Reviewer t2Dn). The key concepts are from the existing literature and the new theoretical results are limited and straightforward.

- Unconvincing experimental validation (Reviewers t2Dn and WDGw). The superiority of the proposed metrics over the many existing ones is not yet convincing. It is not clear how the new concepts can be integrated into existing learning algorithms to obtain better DR methods (Reviewer t2Dn). There is also lack of statistical analysis (Reviewer WDGw) and the authors themselves acknowledged in their rebuttal the difficulty of carrying out such analysis in their current framework.

**Justification For Why Not Higher Score:**

The technical novelty is limited and the experimental validation is weak.

**Justification For Why Not Lower Score:**

N/A

---

### Decision · Program_Chairs · 2024-01-16

Reject